# Calculating with light using a chip-scale all-optical abacus

J. Feldmann[1], M. Stegmaier[1], N. Gruhler[1], C. Ríos[2], H. Bhaskaran [2], C.D. Wright [3] & W.H.P. Pernice[1]

Machines that simultaneously process and store multistate data at one and the same location can provide a new class of fast, powerful and efficient general-purpose computers. We demonstrate the central element of an all-optical calculator, a photonic abacus, which provides multistate compute-and-store operation by integrating functional phase-change materials with nanophotonic chips. With picosecond optical pulses we perform the fundamental arithmetic operations of addition, subtraction, multiplication, and division, including a carryover into multiple cells. This basic processing unit is embedded into a scalable phase-change photonic network and addressed optically through a two-pulse random access scheme. Our framework provides first steps towards light-based non-von Neumann arithmetic.

[1] Institute of Physics, University of Muenster, Heisenbergstr. 11, 48149 Muenster Germany. [2] Department of Materials, University of Oxford, Parks Road, Oxford OX1 3PH, UK. [3] Department of Engineering, University of Exeter, Exeter EX4 QF, UK. J. Feldmann and M. Stegmaier contributed equally to the work. Correspondence and requests for materials should be addressed to W.H.P. (email: wolfram.pernice@uni-muenster.de)

Everyone is familiar with the abacus, invented between 2700 and 2300 BC and one of the earliest mathematical tools[1]. In its most widely known form, the abacus consists of a frame, rods (or wires), and beads to implement a mechanical multistate machine. Each rod represents a different place value (ones, tens, hundreds, and so on), while each bead represents a single digit. By sliding the beads along the rods in appropriate ways, all the basic arithmetic functions of addition, subtraction, multiplication, and division can be carried out, along with even more complex operations. At the same time, the abacus stores the result of such calculations in the (final) position of its beads. In essence, the abacus provides two of the most basic functions of a computer, namely processing (calculation) and memory (storage), and it does this simultaneously and in a single device (or, as an alternative description, at one and the same location). Modern computer systems however, based as they are on the so-called von Neumann architecture, separate, in time and space, the operations of processing and memory. Processing is carried out in the central processing unit (CPU), while separate memory devices store the results of any calculations carried out by the CPU. The constant transfer of data between CPU and memory leads to a 'bottleneck' in terms of the overall speed of operation (the well-known von Neumann bottleneck) and wastes very significant amounts of energy. Computer architectures that can somehow fuse together the two basic tasks of processing and memory (i.e., non-von Neumann architectures) therefore offer tantalizing potential improvements in terms of speed and power consumption. The search for such new computing approaches has been boosted by the advent of so-called memristive devices, i.e., devices that can be excited into multiple (non-volatile) states and whose current state depends on their past history[2–4]. Indeed, such memristive devices can both process and store data simultaneously, and have led to the new concept of multistate mem-processor or memcomputer machines that compute with and in memory[5, 6]. These new approaches to computation provide not only the same computational power as a universal Turing machine (describing all conventional digital computers), but also a range of additional and attractive properties including intrinsic parallelism, learning, and adaptive capabilities and, of course, the simultaneous execution of processing and storage that removes the need for continual transfers of data between a CPU and external memory[5]. Computer architectures based on the multistate compute-and-store operation of a simple abacus can also provide us with such a radically new approach to computing, and one that can work directly in high-order bases rather than just binary. Carrying out such a radical approach entirely in the optical domain using integrated chip-scale photonics would allow for exploiting the ultra-fast signaling and ultra-high bandwidth capabilities intrinsic to light[7].

In this work, we demonstrate a key component in this quest, namely an integrated all-optical abacus-like arithmetic calculating unit. Our approach is based on the progressive crystallization of nanoscale phase-change materials (PCMs) embedded with nanophotonic waveguides. PCMs have been the subject of intense research and development in recent decades, mainly in the context of re-writable optical disks and non-volatile electronic memories[8–10]. A key feature for such applications is the high contrast in both the electrical (resistivity) and optical (refractive

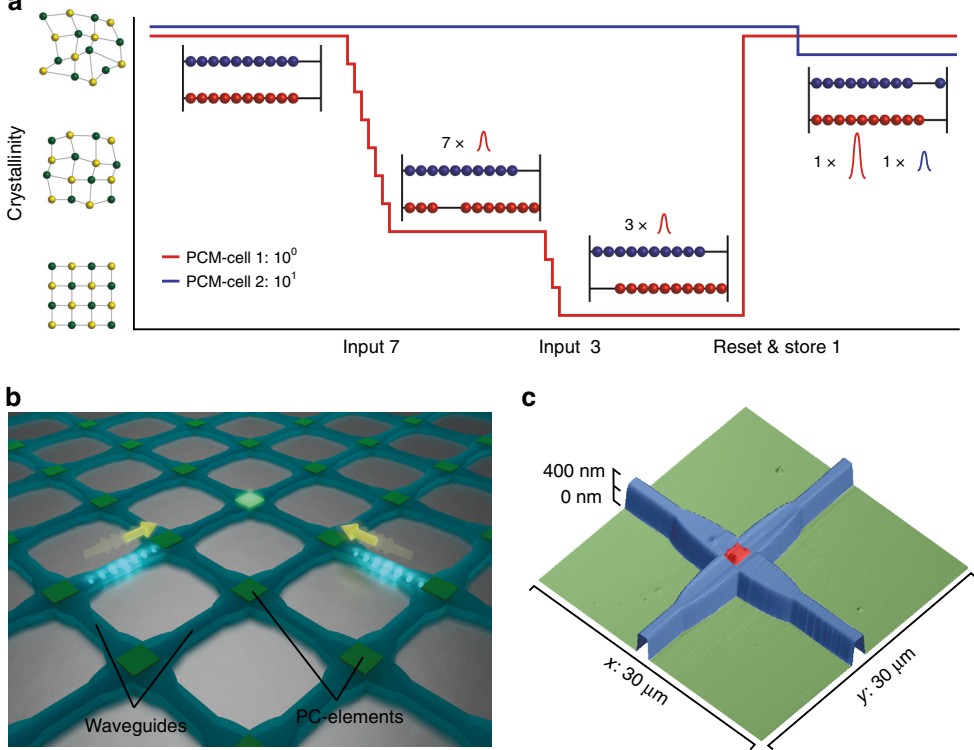

**Fig. 1** All-optical arithmetic using phase-change nanophotonics. **a** Schematic of the operation principle of the photonic abacus. Two phase-change cells are switched with optical pulses in several steps from amorphous to crystalline, each level representing a digit in the decimal system. The red line corresponds to the level of crystallinity of the first PCM-cell (representing ones) and the blue line to the second PCM-cell (representing tens). A reset to the initial state is induced by a higher power pulse. Applying the same crystallization pulses again, all intermediate levels can reproducibly be accessed. The example illustrates the addition '7 + 3 = 10', including carryover to a second phase-change cell. **b** Sketch of a waveguide crossing array illustrating the two-pulse addressing of individual phase-change cells. Only overlapping pulses provide sufficient power to switch a desired PCM-cell. **c** Atomic force microscopy image of a single crossing, with a footprint of 17 × 17 μm

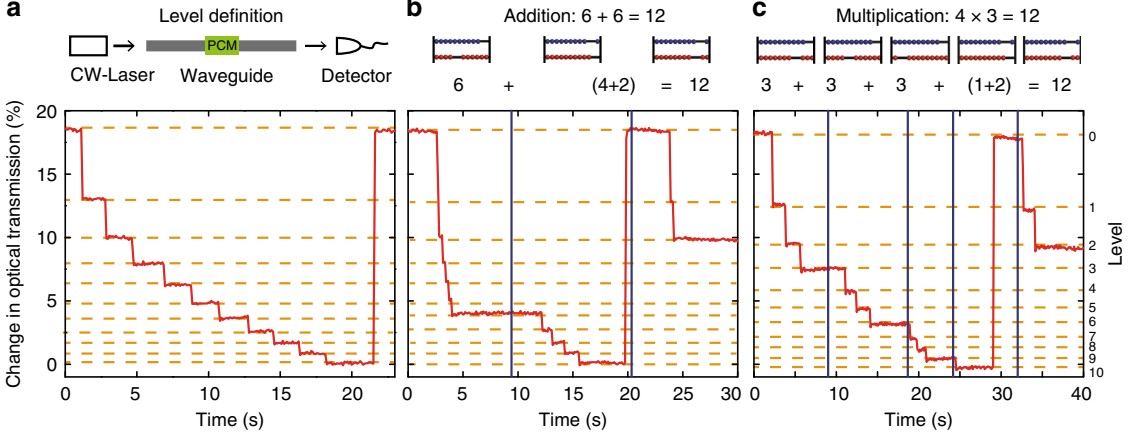

**Fig. 2** Elementary arithmetic operations in base ten. **a** Level definition: Before using a phase-change cell, the levels for the chosen base of operation have to be defined. Therefore the pulse energies are set in such a way that clearly distinguishable and repeatedly accessible levels are obtained. In this example each step downwards consists of a group of five picosecond pulses, the reset pulse of a group of ten ps pulses. **b** Addition: with the transmission levels defined, '6 + 6 = 12' is calculated. By analogy to the operation of an abacus, a carryover is performed when the tenth level is reached and the cell is reset to its initial state. The result '12' is obtained from one carryover and the final state of the PCM-cell, which is at level 2. **c** Multiplication: '4 × 3 = 12', computed applying sequential addition

index) properties of PCMs between their amorphous and crystalline states[8–10]. The high refractive index contrast means that if we place PCMs onto nanophotonic waveguides, we can switch them between states using optical pulses sent down the waveguide, and readout the resulting state optically too. Previous work has used such an approach to demonstrate integrated all-optical memories[11, 12]. Here we show that processing and storage is in fact possible, demonstrating an on-chip abacus-like photonic device that simultaneously combines calculation and memory. We carry out base-10 additions and subtractions (including carryover) in a single PCM-cell using picosecond optical pulses and energies in the sub-nano-Joule range. We also demonstrate successful random user-selective access to each PCM-cell in a two-dimensional array. Moreover, our approach is scalable and could provide photonic integrated circuits with devices that are reconfigurable, namely can be operated as memory, switches, and calculation units, perhaps providing the first steps towards the optical equivalent of electronic field-programmable gate arrays (FPGAs).

## Results

**A phase-change material nanophotonic abacus.** In our all-optical approach which mimics the central element of a non-von Neumann arithmetic calculator, the abacus beads are represented by 'quanta' of crystallization in a phase-change material (PCM) cell[8, 13]. Sliding of a bead to the left/right is thus represented by stepwise amorphization/crystallization[14] which can be triggered by nanosecond, picosecond or even femtosecond optical pulses[15–17]. Due to the large refractive index contrast between the amorphous and crystalline phase of PCMs[18], the crystallinity and thus the stored quanta can be conveniently read-out by optical means. This concept is sketched in Fig. 1a which illustrates the addition '7 + 3 = 10' as a simple example of a calculation in base ten with carryover. Two phase-change cells are used to represent the first (depicted red) and second (depicted blue) digit of the used base-10 system used. Initially, both cells are in the amorphous phase which represents the number 0. In the first step, as many recrystallization steps as the first summand, seven in this case, are initiated in the red PCM-cell by appropriately designed optical pulses. Subsequently, recrystallization steps equivalent to the second summand, 3, are carried out which eventually result in full crystallization. In analogy to the abacus where at this point all the

red beads are shifted back to the left and one blue bead is slid to the right (a carryover), the red PCM-cell is now amorphized (by a single more intense optical pulse) and in the blue PCM-cell one crystallization step is performed. Arithmetic calculations and data storage are therefore carried out simultaneously in the self-same PCM-cell. This allows us to avoid the well-known von Neumann bottleneck that plagues conventional computers, in which data has to be continually transferred between memory and CPU during and after a calculation[19, 20].

We implement the abovementioned photonic abacus out of phase-change devices[12, 21] integrated into the chip-scale photonic networks sketched in Fig. 1b. Integrated optics provides a scalable platform to implement a fast and random-access architecture that can also eliminate heating and latency issues associated with electrical interconnects[22, 23]. The rectangular waveguide array with a PCM-cell realized at every waveguide crossing point enables selective addressing and manipulation of each of these basic arithmetic units with the sketched two-pulse switching method (described in more detail below). This crossed-waveguide approach is potentially scalable and thus readily leads to on-chip arrays capable of efficient and powerful computation. The overall size of the array is eventually limited by absorption within the PCM cells, which can be alleviated using multi-pulse addressing schemes. Figure 1c shows an atomic force microscopy image of a single waveguide crossing with its integral PCM-cell on top (red). We operate our on-chip photonic circuits at telecom C-band wavelengths (1530–1565 nm) and use picosecond optical pulses, that here for simplicity are electronically generated off-chip, to switch the PCM-cells. We note that in principle the pulse routing and generation could also be realized on-chip using heterogeneous integration with light sources and through combination with active elements available in photonic foundries[24, 25]. Details on the device design and the experimental set-up are given in methods section. Using this platform, we first demonstrate the operation of one basic arithmetic unit, including a carryover. We then proceed to the operation of multiple units within an array of PCM-cells, illustrating photonic random memory access.

**All-optical arithmetic processing.** As shown in Fig. 2a, the single processing unit comprises a straight waveguide with a phase-change cell of 7 μm length on top (in this first demonstration we use AgInSbTe (AIST)). Due to the substantial refractive index

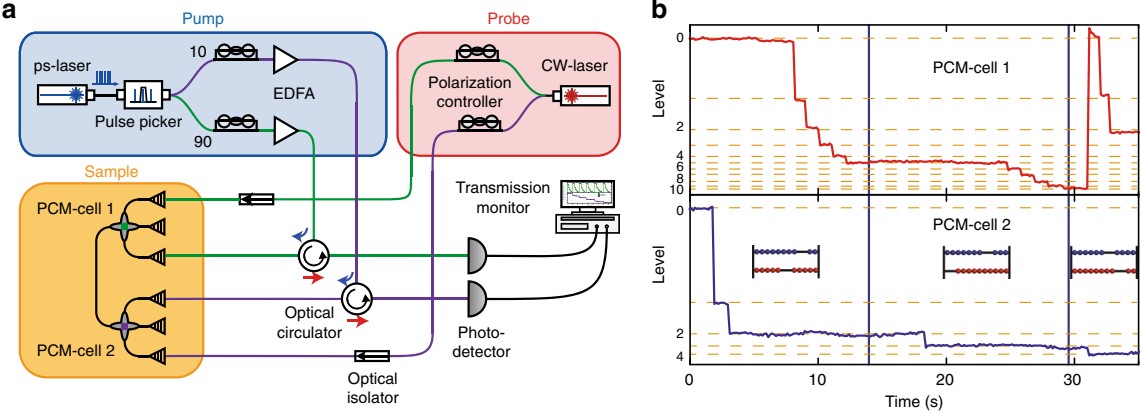

**Fig. 3** Two-digit arithmetic with carryover. **a** Pump-probe set-up using two separate phase-change cells. The pump pulse to switch 'PCM-cell 1' is split (90:10) and partially sent to the second PCM-cell. When a high energy reset pulse is sent to the first cell in order to perform the carryover, the energy of the pulse guided to the second cell is enough to induce a crystallization step in the second cell. **b** Calculation of '74 − 17 = 57' using two PCM-cells and the nine's complement method for subtraction, which is equal to the addition '25 + 17 = 42'

contrast between the amorphous and crystalline state[18], the optical absorption (and thus optical transmission) for light traveling down the waveguide depends sensitively on the phase-state of the cell[11]. This is demonstrated in Fig. 2a where the transmission through such a waveguide is plotted as the PCM-cell is taken through a full switching cycle. Starting from the amorphized state (low absorption, thus high transmission), stepwise crystallization is carried out with identical picosecond pulses with pulse energies of 12 pJ each. In the presented operation, five consecutive pulses are used for every crystallization step. Subsequently, the device is reset (reamorphization) with ten 19 pJ pulses. Single pulse mode is also possible, resulting in faster and lower energy operation, but with lower optical contrast (Supplementary Note 3).

The first all-optical arithmetic operation we demonstrate is base-10 addition of '6 + 6', cf. Fig. 2b. Starting from state 0, pulse sequences equivalent to the first summand are sent into the waveguide which sets the PCM-cell to level six. Then the second summand is added by sending in the corresponding pulses. When reaching the tenth level, the PCM-cell is reset to level 0 before the rest of the input sequence is applied. While resetting the cell, one pulse sequence is sent to a second PCM-cell representing the next highest order multiple of the base to store the carryover information (not shown in the graph). At the end of the calculation, the shown PCM-cell is at level 2 while the second PCM-cell is at level 1, revealing the expected answer of '12'. Next, in Fig. 2c we demonstrate multiplication of '4 × 3'. Since this operation can be implemented by successive addition, the scheme is analog to the already presented procedure. As expected, the final states of the PCM-cells after 4 successive pulse sequences and one carryover are '2' and '1', thus representing the correct result '12'.

As mentioned above, the carryover is stored in a second PCM-cell and arithmetic operations with more than a single digit can be performed using multiple cells and exploiting the additional computational power and efficiency that comes with operating directly in base ten. Each cell represents a single place value (ones, tens, hundreds, and so on) corresponding to the different rods of an abacus. Figure 3a shows the experimental pump-probe set-up used for two-digit operations including the carryover to the second PCM cell. To detect the cell-states a continuous wave probe laser is used and the optical transmission through the PCM-cell is monitored with a photodetector. As in the previous experiments (above), the pump pulse to switch the cells is picked from a picosecond-laser. The pulse which was formerly used to

operate the first phase-change cell is now split in two (here off-chip, but this could in the future also be carried out on-chip, for example with a Y-splitter) and guided to both PCM-cells. If the splitting ratio is adjusted in an appropriate way (which we accomplish by using optical amplifiers), the part of the energy of the crystallization pulse for the first element which is sent to the second cell does not induce a phase transition. In contrast, if the high energy amorphization pulse is sent to the first element, a crystallization step is induced in parallel in the second cell and thus it directly stores a carryover. Figure 3b now shows a subtraction example carried out directly in base-10 and with two digits (see Supplementary Notes 5, 6 for additional information). Here we use the numbers complements approach which requires addition routines only[26]. We subtract '17' from '74' (which of course should yield the answer '57') by adding '17' to the nine's complement of the minuend '74', which is '25'. In a first step, we set the state of the two PCM-cells corresponding to the minuend. Because we use the nine's complement we switch the first cell (ones) to '5' and the second cell (tens) to '2', so that the phase states of the two elements now hold the value '25'. Now the subtrahend '17' is added, consisting of a single crystallization pulse in the second cell and seven pulses sent to the first cell. After five of these seven pulses, the 'full' crystalline level in the first cell is reached and storage of the carryover is performed. The reset pulse for cell one is now utilized to induce a crystallization step in the second cell (representing the tens), as described above. After sending all the required pulses, as above, the phase states of the PCM-cells represent the value '42' which is the nine's complement of '57', the correct result of the subtraction. In this example only single pulses per step with energies of 10 pJ for crystallization and 20 pJ for amorphization were used, showing that in total 15 steps were needed for the calculation with an accumulated calculation time of less than 15 ns, assuming that a single switching event for picosecond pulses takes place in less than a nanosecond as shown in ref. [12]. The speed of operation is only limited by the crystallization and cooling time of the phase-change material and approaches the GHz-regime when using picosecond pulses. In our experiments, the pulses were generated with an electronically triggered pulse picker and were sent, for ease of demonstration, manually to the photonic abacus device. This led to artificially long overall operation times (for the completion of calculations) of the order of seconds. However, we note that the ancillary circuit needed to perform the necessary signal generation, as well as the detection of the carryover, can be envisaged and implemented on chip as well in future work

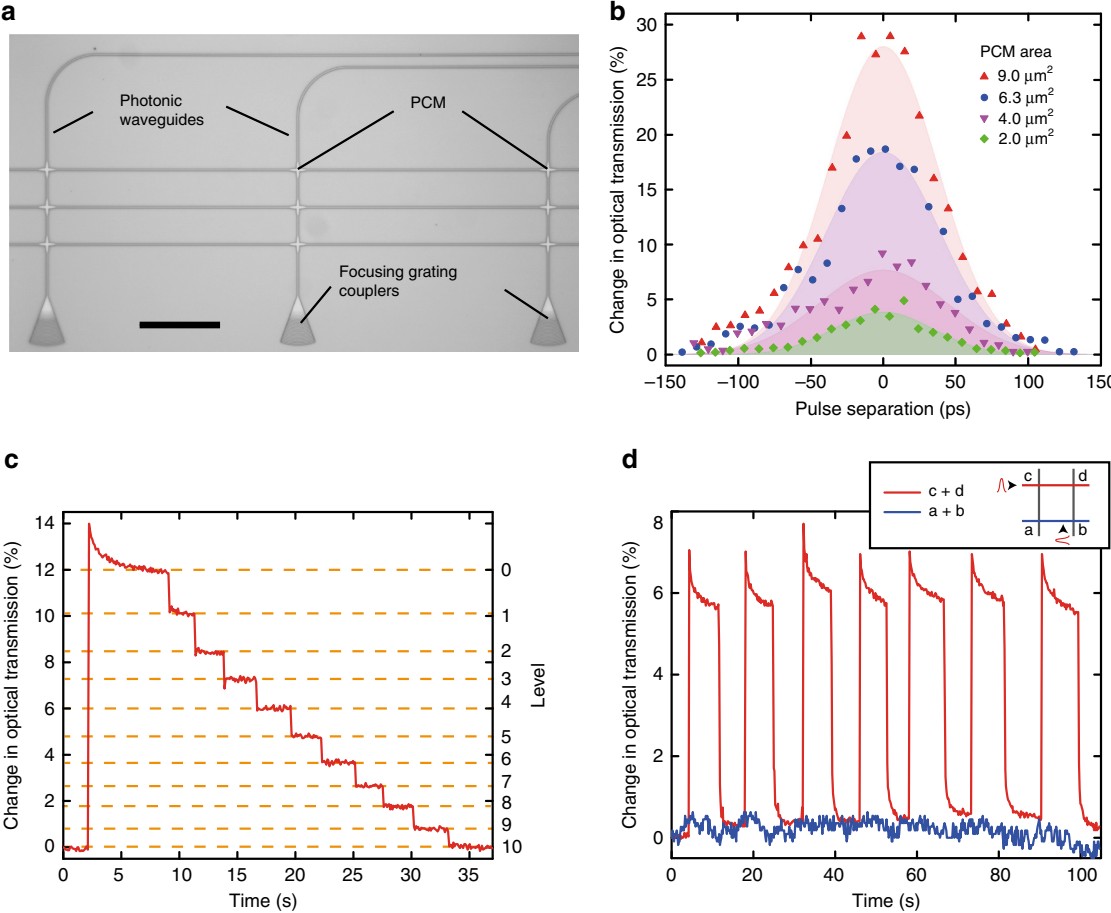

**Fig. 4** Two-pulse switching of waveguide crossings. **a** Optical micrograph of a studied crossed-waveguide photonic array (Scale bar is 100 μm). **b** Measured change in optical device transmission upon amorphization for different relative arrival times of the two 1 ps pulses. Since the pulse powers are chosen low enough, separate excitations (delay more than 150 ps) cannot trigger a transition. Only simultaneously arriving pulses at the PCM-cell induce a phase-change. **c** Multilevel operation of a single crossing shows that also the arithmetic operations presented in Fig. 2 can be executed with the two-pulse switching scheme. **d** Transmission across cells 'c' and 'd' (red) and 'a' and 'b' (blue) during repeated switching of cell 'd' via pulses sent along 'cd' and 'bd' (cf. inset). The shown transmission through the 'a' and 'b' cells is unaffected by the switching, which demonstrates that only the selected cell 'd' is manipulated

to exploit the speedup potential of the PCM-based photonic abacus.

Assuming that division is repeated subtraction, all four basic arithmetic operations can thus be executed with our on-chip device. As shown in Supplementary Note 4, we can also use other number bases should this be desirable and/or suited to a particular problem.

**Two-pulse switching of individual phase-change photonic cells**. So far, we have demonstrated arithmetic operations with individual PCM-cells, each representing a single abacus slide. In the following, we take advantage of nanophotonic circuits to construct a scalable, multi-cell architecture, representing a multi-slide abacus, for fast, random-access optical operation. For this purpose, on-chip photonic networks[27] are ideal since they enable fast on-chip routing and multiplexing for random-access and high-speed parallel read-out and processing[22, 23]. Here, we use an array of low-loss and low-crosstalk waveguide crossings (Supplementary Note 7), as previously shown in Fig. 1b, c and implemented as in Fig. 4a. To guarantee random user-selective access to each PCM-cell in the array, it is important that the pulse energy is transferred only to the desired waveguide crossing region and that other cells in the propagation direction remain unaffected. In this random access scheme, a particular PCM cell can be precisely selected, in contrast to for example NAND-flash memories in which multiple cells have to be rewritten in order to change a single cell[28]. We achieve such a random access of all phase-change cells by a two-pulse addressing technique. The pulses are launched into the photonic array via two orthogonal waveguides, superimposing at the crossing incorporating the PCM-cell that we wish to switch. (Note that in this second set of experiments we use, for convenience, the phase-change material $Ge_2Sb_2Te_5$, but we do not find significant differences between devices made from AIST or GST and thus are able to maintain the same measurement and fabrication approaches with either material, Supplementary Note 2). One pulse is used to provide the row address, while the second pulse yields the column address in our 2D array. Fast on-chip selection of the two waveguides could for example be carried out by wavelength[29] and mode-division[30] multiplexing schemes, e.g., by adjusting the wavelengths and/or modes of the excitation pulses accordingly (note that in this work the selection of the waveguides was realized off-chip and the experimental set-up is shown in Supplementary Fig. 1). Such devices can be readily realized using photonic foundry services[24, 25]. Importantly, we select the pulse energy such that a single pulse alone is not intense enough to switch a cell. Thus, as shown in Fig. 4b, only when both pulses arrive at a particular cell together, the phase-change material heats up sufficiently to induce a phase transition

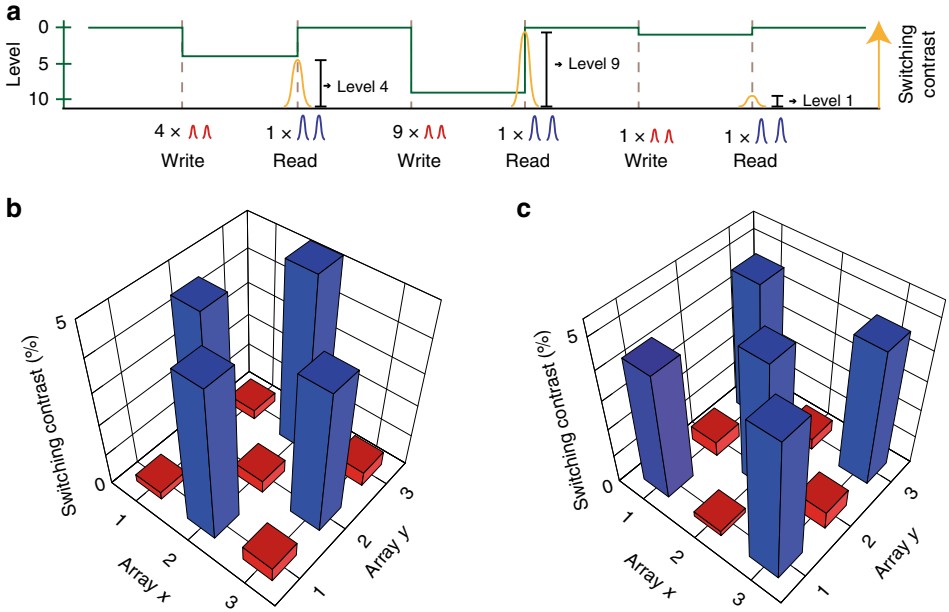

**Fig. 5** Random-access readout in a 3 × 3 crossed-waveguide array. **a** Readout principle for crossing arrays using the two-pulse-technique. When applying a reset pulse to a particular cell, the switching contrast (yellow) corresponds to the initial state of that cell and can therefore be used as a readout mechanism. **b**, **c** Experimental read-out of a 3 × 3 array in two complementary states whose PCM-cells are operated in base-2. While negligible change in optical transmission (red bar) reveals that the cell was set in state 1, an increase of transmission by about 4.5% (blue bar) corresponds to state 0

(essentially corresponding to an optical AND operation). The change in optical transmission is plotted as a function of the delay between the two pulses. If they overlap exactly at the crossing point (delay = 0 ps) the switching contrast is at maximum. If, on the other hand, the delay is more than 150 ps, the contrast is zero. As shown in Fig. 4c, the multilevel (multistate) operation is fully maintained with the two-pulse switching scheme. Thus, the previously demonstrated arithmetic functionality is preserved in the crossed-waveguide photonic array, and such designs portend large-scale all-optical processing units as well as multilevel random access photonic memories[12]. Additional information about the switching contrast is provided in Supplementary Note 8.

**Optical random access with nanophotonic circuits**. Our ability to select (switch) any particular cell, and only that cell, using the crossed waveguide approach is demonstrated in Fig. 4d. We plot the change in transmission for light traveling along a row of a waveguide array while repeatedly switching one of the cells in that row (red line), and simultaneously compare it to the transmission along a second row not containing a switched cell (blue). Only the transmission through the row with the switched cell (see inset in Fig. 4d) changes, while the other cells in the array remain unaffected. To readout the state of any individual cell within the crossed-waveguide array, a simple transmission measurement is no longer appropriate, since the optical transmission along any row or column in the waveguide array will be influenced by all the phase-change cells lying on the same row or column. In Fig. 5a we demonstrate an alternative readout scheme suited to the crossed-waveguide array concept and which makes use of two-pulse switching. When a PCM-cell is set to a specific level and a reset operation consisting of two overlapping pulses is applied, the change in contrast depends on the initial state of the phase-change material. For example, resetting from level four to level zero, as shown in the diagram, reveals a lower change in optical transmission compared with switching from level nine. Thus, by sending in a reset pulse to the crossing of interest and detecting how much the transmission changes, we can selectively read the state of each PCM-cell in the array (though in a destructive

manner, see Supplementary Note 9 for further details). Figure 5b, c shows the results of an experimental implementation of this approach. Here we plot the readout contrast for two different states of the nine phase-change cells. The pulse energies used in this case are 15 pJ for each of the two pulses of a reset operation and around 5 pJ for the respective write operation pulses. The contrast of about 4.5% when switching from the crystalline to the amorphous level is clearly distinguishable from the small changes (of <0.5%) that result from resetting an already amorphous element. While this read-out scheme is destructive, non-destructive array readout schemes based on monitoring the post-excitation transient response of PCMs[31, 32] might be possible. To overcome the destructive readout, a two-pulse scheme could be used to directly rewrite the cell after reading. This way the memory system would keep all the information also after the readout.

## Discussion

Our on-chip abacus-like all-photonic calculator simultaneously combines calculation and memory in a single PCM cell and can operate directly in arbitrary bases using picosecond pulses. Convenient stepwise calculations can be performed using single-optical pulses with the same pulse energies for each step, leading to potential GHz operation speeds. Such multistate compute-and-store capability can set the stage for a new generation of fast and efficient Turing-complete arithmetic processors. The extraordinary properties of PCMs, e.g., stability of the phase-state for years[13], sub-ns crystallization times[33], sub-pJ switching energies[34], and endurance potentially up to $10^{15}$ cycles[35], hold promise for ultra-fast, low-energy all-optical processing. Our concepts are scalable so as to deliver large-scale photonic processing networks, for example by using the crossed-waveguide array approach that we have demonstrated here. In the current work the signal generation and detection of a carryover was achieved off-chip but can be envisaged to be completely integrated on a photonic chip, removing the need for time and energy sapping electrical-optical conversions, as well as leveraging the significant benefits of light-based systems, such as ultra-fast, ultra-high bandwidth, and low-energy signaling. Taken in tandem

with the rapid progress being made in the silicon photonics field, our work presages a new generation of light-based arithmetic processors.

## Methods

**Device fabrication**. The phase-change photonic devices presented in this work are fabricated using a three-step electron-beam (e-beam) lithography (EBL) procedure with a 50 kV EBL system (JEOL 5300). In the first step, alignment markers are deposited on top of a commercially purchased silicon wafer (Rogue Valley Microdevices) with a 330 nm $Si_3N_4$, 3334 nm fused silica layer stack. The marker areas are defined by EBL in the positive tone e-beam resist Polymethylmethacrylat (PMMA). After 2 min development in 1:3 MIBK:Isopropanol, ~5 nm chromium and ~120 nm gold are e-beam evaporated to achieve good contrast for high-precision alignment. The PMMA is then removed by lift-off in acetone. In the second step, photonic circuitry is defined in the negative-tone e-beam resist ma-N 2403. The resist is developed in MF-319 for 60 s and then placed on a hotplate at 110 °C for 2 min to reduce the surface roughness. Afterwards, the mask pattern is transferred into the silicon nitride layer by reactive ion etching in CHF3/O2 plasma and the remaining resist is removed by oxygen plasma subsequently. In all devices the silicon nitride layer is fully etched down to the underlying silicon dioxide. Finally, masking windows for the PCM-cells are opened in another PMMA-based EBL-step. Then 10 nm of PCM are sputter deposited and subsequently capped with another 10 nm indium tin oxide (ITO) film to prevent oxidation. The deposition was carried out with a Nordiko sputter system from commercially purchased targets. The GST (2.5″ solid target purchased from Super Conductor Materials, USA) was deposited at 30 W DC with a rate of approximately 3.6 nm/min, the AIST (2.5″ target purchased from Super Conductor Materials, USA) at 30 W DC with approximately 3.4 nm/min and the ITO (3″ target purchased from Test-bourne, UK) at 120 W DC with ~11 nm/min. Finally, the PMMA is removed in another aceton-based lift-off and the PCMs are crystallized by placing the chip on a hotplate at 250 °C for more than 10 min.

The on-chip optical devices are fabricated with a waveguide width of 1.2 µm and therefore operated in single mode at a wavelength of 1550 nm. Focusing grating couplers are used as input and output ports for transmission and switching experiments.

**Measurement set-up**. The set-up used to switch and characterize the phase-change photonic circuits is shown in Supplementary Fig. 1 and consists of two optical paths. The first path contains the probe light (colored red) which is used to perform transmission measurements and therefore allows us to monitor the phase state of a PCM-cell. Light from a continuous wave (CW) laser (Santec, TSL 510) is coupled into the on-chip waveguide via focusing grating couplers inscribed at the ends of each waveguide. Before coupling into the waveguide the polarization is adjusted to match the desired mode profile within the on-chip waveguide. After passing the on-chip device, the transmitted light is detected by a low-noise photodetector (New Focus, Model 2011).

The second measurement path is the one for the pump light (colored blue) and is used to generate optical pulses to switch the PCM-cells. A pulse picker, consisting of an acousto-optic modulator (Gooch and Housego, Fibre-Q T-M200) controlled by an electrical pulse generator (Agilent, HP 8131 A) and a computer is used to select optical pulses on demand from a modelocked picosecond laser (Pritel, FFL Series) working at a repetition rate of 40 MHz with a pulse length of about 1 ps. Behind the pulse picker, the selected pulses are split by a 50:50 beam splitter and separately amplified by low-noise erbium-doped amplifiers (Pritel, LNHPFA-33). To control the time delay at which the pulses arrive at a PCM-cell an optical delay line (OZ Optics, ODL-300) is inserted into one optical path. Separation of the pump pulses and the probe light for independent off-chip detection is achieved by means of optical circulators which are used to guide pump and probe light to different detectors. The picosecond pulses are recorded with a 12 GHz photodetector (New Focus, Model 1554-B) and a high bandwidth 6 GHz oscilloscope (Agilent infiniium, 54855 A). This allows us to measure the time delay between the two pump pulses and thus to match their arrival time at a PCM-cell in the two-pulse switching scheme described in the main text.

Prior to arithmetic operations, the PCM-cell is conditioned by repeating the switching cycle several times to achieve good reproducibility, as already described in previous work[12].

**Data availability**. All relevant data are available from the authors.

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

## Acknowledgements

Additional data supporting the conclusions are available in Supplementary Materials. The authors acknowledge support by Deutsche Forschungsgemeinschaft (DFG) grants PE 1832/2-1 and EPSRC grant EP/J018783/1. M.S. acknowledges support from the Karlsruhe School of Optics and Photonics (KSOP) and the Stiftung der Deutschen Wirtschaft (sdw). C.R. is grateful to JEOL UK and the Clarendon Fund for funding his graduate studies. H.B. acknowledges support from the John Fell Fund and the EPSRC (EP/J00541X/2 and EP/J018694/1). The authors also acknowledge support from the DFG and the State of Baden-Württemberg through the DFG-Center for Functional Nanostructures (CFN). The authors thank S. Diewald for assistance with device fabrication.

## Author contributions

W.H.P.P., H.B., and C.D.W. conceived the experiments. N.G. and C.R. fabricated the samples with the help of M.S. and J.F. J.F. and M.S. performed the measurements. All authors discussed the results and wrote the manuscript.

## Additional information

**Competing interests:** The authors declare no competing financial interests.

