## [Peer Review File · Nature Communications]

Reviewers' comments:

Reviewer #1 (Remarks to the Author):

NCOMMS-17-06101 claims in title and conclusion that the framework and the realizations are the first steps to all-optical computing. The paper is organized by two parts. The first part is multistate operation of single optical phase-changed material (PCM) cell which was regarded as (or emulated) optical abacus. In the second part, PCM cells placed at waveguide crossings of were selectively written and reset by two-pulse addressing and such operations were regarded as random access. Despite the repeating descriptions of picosecond pulse injection, the arithmetic operations reported in the paper (Figs. 2, S2, S3, and S5) eventually needed total operation times in the order of several ten seconds (~ 20 s for adding '10'), which were much longer than corresponding mental calculation time (1 s or less) a human (brain) needs. Certain prospects for significant acceleration of calculations as it can compete with current computers are not presented.

Such technical journals that have the scope of timely publication of leading technologies for future devices/systems must not focus on the paper like this. But I think that the topic of optical abacus may be appropriate for Nature Communications.

Unfortunately, as a paper of optical abacus, current paper is full of inappropriate and/or misleading claims. The paper should not be published in its present form for not only Nature Communications but also (Nature) Scientific Reports.

Followings are the criticisms for the main claims.

(1) The authors should reconsider what the abacus is. An abacus is an useful tool, but it is just a tool for helping man's mental calculation and it never be a computer or a processor. The beads are always operated by man's fingers and the operations are not automated. In what was presented here, the carryover operation was not automated. A human judged whether the carryover/reset was needed and injected a reset pulse. Such an intentional operation is no problem as an abacus, but their PCM cell does not work as a computer/processor with reset/carryover. It is obvious that an abacus cannot be operated faster than mental calculation. However, if an external processing unit (CPU) is employed to take over the human, the abacus would be no longer needed.

They should remind that in abacus many beads can be moved at the same time in one step (e.g. in $1+8=9$, 8 beads can be moved in one step) and in operation of 2 digits ($20+14$) every digit can be operated separately in one step ($2+1$ and $0+4$). On the contrary, unlike the normal abacus, in the Figs. mentioned above their optical abacus need N step increment operations (by 1) for addition/subtraction ($6+6:N=12$, $5+6:N=11$, $3+9:N=12$, $20+14:N=34$) and multiplication ($4*3:N=12$, $4*2:N=8$). Do the authors believe that even in ancient days they prefer such the abacus that can essentially do increment by 1 only? As operand become larger (e.g. $100+100$, $1000*1000$, ...) their abacus becomes more inefficient and mental or writing calculations becomes much efficient. Do they really think that the operations in optical abacus presented here is "wise", "cool" or "smart?" Otherwise I believe that such an inefficient calculation machine is not appropriate for Nature Communications.

(2) Their PCM cells are not automated for multi-digit calculation and carryover and therefore do not work as computer/processor. They did not report computing at all and so the work/framework should not be compared with computer and/or computer architectures. The current title is invalid. Their framework is not related with non-von Neumann computer architectures.

(3) The paper claims that PCM cells successfully worked as optical abacus. But I think that the demonstration is at least insufficient. It is obvious that $M*N$ multiplication is emulated by $M*N$ times increment by 1, which should not be an implementation of multiplication for a processing unit. As for subtraction, conversion between decimal number and 9's complement is needed. How do they implement the conversion on their PCM cells? It seems that implementation of subtraction ($9-7$ and $9-9$ here) seems difficult for their PCS cells. For computation, how the machine recognizes that which PCM cell records decimal value or its 9's complement. Moreover, even in addition, addend N is replaced by N times addition of 1. After all, what they demonstrated here was just an optical counter as described in Section 5 of Supplementary materials. The fact that $10+10$ ($10*2$) were emulated by $(1+1+1+1+1+1+1+1+1+1+1+1+1+1+1+1+1+1+1+1)$ seems funny. I don't think that this is the demonstration of multistate compute operation. As pointed,

such an optical counter is much inefficient than mental calculation. I believe that their current implementation spoils merits of multistate operation and multistate recording. More efficient implementation should be presented as 2-digit addition, subtraction, and multiplication operations can be operated by a few steps.

(4) The second part reports addressing and random access to arrayed PCM cells. Although abacus is not a computer, such technology may be needed for multi-digit calculation. However, in Fig. 4, 3*3 PCM array was operated in base-2 (binary) only. Compatibility with the base-10 (decimal) optical abacus in the first part is not demonstrated. In the write (switch) operation (Fig. 3D) it was ambiguous that it was compatible with base-10.

The author may continue efforts toward acceptance of this work. If they wish to publish the work as optical abacus, addressing the comments (1)-(3) and excluding the second part is my recommendation. Instead of the second part, they should add theoretical discussion about how their optical abacus can be operated much faster than human's mental calculation without the abacus (~millisecond or faster.) Or they may add experimental result of arithmetic operation with total time in millisecond (although multi-step structure may be ambiguous or noisy.) Unlike the optical memory, speed is highly important for a calculator and as a Nature Communications paper such high impact is needed.

Otherwise, if they wish to publish the work as a prototype of all-optical computation, they should withdraw the claim of optical abacus and the topic should be changed to such as all-optical programmable logic device. It may be possible by excluding the not-automated operations (reset/carryover.)

I describe other comments below.

(5) In Figs. 2, and S3, the signal of the PCM-cell 2 (first digit/blue bead) should be plotted as in S5, otherwise the claim of carryover and 2-digit calculation were false and invalid.

(6) Fig. 2B and 2C report very simple 1-digit calculation ($6+6$, $4*3$) which the readers may not feel that such optical abacus is useful. It is also important to demonstrate that the PCM cell was worked not just an optical counter. I recommend the addition and multiplication operations by 2-digits (e.g. $26+35$, $16*5$) and direct arithmetic operation in the first digit (e.g. $2+3$.) If possible, direct change between two multi-level states such as $0>6>0$ corresponding to sliding many beads can be used and be good.

(7) According to the text, the addressing of the PCM cells in the waveguide array was achieved by sophisticated techniques combining two-pulse launching, wavelength-division multiplexing and mode-division multiplexing. But detail experimental conditions for Figs. 3 and 4 are not disclosed. More detailed clarifications should be made as someone can reproduce/simulate the experiment.

(8) The random access readout reported in Fig. 4 reset the referred cell to 0. After the random readout, some cells are reset to 0 and other cells hold the data. For instance, 199999999 is changed to 190999999 after the random readout of 2 cells. Do they think such random readout scheme is actually useful?

(9) The use of following words should be clarified or justified: non-von Neumann computer architectures, scalable, chip-scale, telecom C-band, Turing-complete computers. What they mean random access is ambiguous.

(10) Fluctuation of the expressions such as abacus/counter, abacus/processor, and abacus/memory makes the paper confusing. They should revise the paper as is more coherent.

Reviewer #2 (Remarks to the Author):

I think there is some nice technical work described in this manuscript, in particular the use of PCM cells organized in arrays. So that should be published somewhere.

However, in my opinion the article is somewhat misleading. The title "Computing with light using a chip-scale all-optical abacus", suggests to me that all the arithmetic operations are performed

optically. Reading the article and supplementary material though, I get the impression that when a carry-over occurs then a reset pulse must be sent to one digit, and an increment by one pulse sent to the next higher digit. These pulses appear to be generated electronically in the experimental setup.

For the system to be truly all optical there needs to be all optical switching of light pulses by the PCM cells. So that whenever a carry over needs to occur then the optical chip itself can sort out what needs to be done.

Since all-optical switching does not appear to occur on the chip presented here I feel that the work does not represent a really significant advance in optical computing that would be needed to warrant publication in Nature Communications.

Reviewer #3 (Remarks to the Author):

This is an interesting paper, and I would recommend publication in Nature communications after major revision. This paper is built on and an improvement upon the work done previously by the authors. It is effectively a combination of the devices and concepts presented in:

- Integrated all-photonics non-volatile multi-level memory Nat. Photonics 9, 725-732 (2015) and
- Arithmetic and biological inspired computing using phase change materials Adv. Mater. 2011, 23, 3408-3413

Abacus application is weak. I fail to see how this is in anyway an efficient or sensible way of arithmetic computing. In the least the authors should put the abacus in context with alternative technologies and discuss what this adds to the field.

The novelty of the manuscript lies in the positive steps made in integrated all-optical computing, the manuscript effectively implements a random access memory integrated photonic cross-bar architecture reminiscent of the recent electronic cross-bar architectures. Operating at telecom c-band wavelengths with ps optical pulses also makes it industrially friendly and comparable to say electronic DRAM speeds. An improved novel two pulse random access scheme is used allowing access to any cell in a phase change array. One major shortfall is that the read operation complementing the two pulse random access scheme is destructive. Effectively making the system a Read-once memory system.

In the first half of the manuscript AIST is used as the phase change material. For the random access section the material is changed to GST. Authors should provide the reasoning for this change in the manuscript. How did the pulsing parameters (energy, width) change when using GST? Was AIST not good for random access?

Some points that need to be addressed:

- The authors provide results for random access operation for up to a 3 x 3 array. How does this scale? How does their pulse energies scale with a growing array. A discussion on scaling should be provided.
- Nothing on endurance cycling is provided. How many times do they switch the cells before they fail. Also any difference between single pulse and two pulse scheme in terms of endurance of the cells. As the cells need to be destructively read in the case of the two pulse scheme, this must reduce their endurance significantly?
- I would suggest to move the discussion of "storing the carryover" from supporting info to main text as the splitting of the signal and utilising a high power reset pulses to partially crystallise the second pcm cell was not clear at all in the main text.
- Annotation for figure 1b would be useful to the reader.

Calculating with light using a chip-scale all-optical abacus

J. Feldmann, M. Stegmaier, N. Gruhler, C. Ríos, H. Bhaskaran, C.D. Wright, W.H.P. Pernice

Response to reviewers

We thank all the referees for their insightful reviews of our manuscript and their very helpful recommendations on how to improve it. We have followed such recommendations in full, as described in detail below.

Response to reviewer 1:

NCOMMS-17-06101 claims in title and conclusion that the framework and the realizations are the first steps to all-optical computing. The paper is organized by two parts. The first part is multistate operation of single optical phase-changed material (PCM) cell which was regarded as (or emulated) optical abacus. In the second part, PCM cells placed at waveguide crossings of were selectively written and reset by two-pulse addressing and such operations were regarded as random access. Despite the repeating descriptions of picosecond pulse injection, the arithmetic operations reported in the paper (Figs. 2, S2, S3, and S5) eventually needed total operation times in the order of several ten seconds (~ 20 s for adding '10'), which were much longer than corresponding mental calculation time (1 s or less) a human (brain) needs. Certain prospects for significant acceleration of calculations as it can compete with current computers are not presented.

Our Response: The arithmetic examples shown in Figs. 2, S2, S3 and S5 are indeed executed in total operation times of several tens of seconds. We note, however, that the longer run times were employed only for demonstration purposes to illustrate the pulse sequences used to perform a certain arithmetic task. The waiting time between the individual pulses (which in the examples shown in the main text leads to long operation times) is in a real application only limited by the crystallization and cooling time of the PCM-cell, which lies in the sub-ns range when switching with picosecond pulses [see M. Stegmaier et al., "Nonvolatile All-Optical 1×2 Switch for Chipscale Photonic Networks", *Advanced Optical Materials* 5, 1600346 (2017)]. Therefore, our all optical devices can (as a lower bound) at least be operated with GHz frequencies, which is comparable to modern CPUs (but unlike modern CPUs, our approach calculates *and* stores simultaneously in the same cell, and can carry out calculations directly in high-order bases). In the revised manuscript, the main text was edited in order to point out the speed limiting factors more clearly (page 6):

"In this example only single pulses per step with energies of 10 pJ for crystallization and 20 pJ for amorhization were used, showing that in total 15 steps were needed for the calculation with an accumulated calculation time of less than 15 ns, assuming that a single switching event for picosecond pulses takes place in less than a nanosecond as shown in [15]. The speed of operation is only limited by the crystallization and cooling time of the phase-change material and approaches the GHz-regime

when using picosecond pulses. [...]In our experiments, the pulses were generated with an electronically triggered pulse picker and were sent, for ease of demonstration, manually to the optical abacus device. This led to artificially long overall operation times (for the completion of calculations) of the order of seconds. However, we note that the ancillary circuit needed to perform the necessary signal generation, as well as the detection of the carry-over, can be envisaged and implemented on chip as well in future work to exploit the speedup potential of the PCM-based optical abacus.”

Followings are the criticisms for the main claims.

(1) The authors should reconsider what the abacus is. An abacus is an useful tool, but it is just a tool for helping man’s mental calculation and it never be a computer or a processor. The beads are always operated by man’s fingers and the operations are not automated. In what was presented here, the carryover operation was not automated. A human judged whether the carryover/reset was needed and injected a reset pulse. Such an intentional operation is no problem as an abacus, but their PCM cell does not work as a computer/processor with reset/carryover. It is obvious that an abacus cannot be operated faster than mental calculation. However, if an external processing unit (CPU) is employed to take over the human, the abacus would be no longer needed.

They should remind that in abacus many beads can be moved at the same time in one step (e.g. in $1+8=9$, 8 beads can be moved in one step) and in operation of 2 digits ($20+14$) every digit can be operated separately in one step ($2+1$ and $0+4$). On the contrary, unlike the normal abacus, in the Figs. mentioned above their optical abacus need N step increment operations (by 1) for addition/subtraction ($6+6:N=12$, $5+6:N=11$, $3+9:N=12$, $20+14:N=34$) and multiplication ($4*3:N=12$, $4*2:N=8$). Do the authors believe that even in ancient days they prefer such the abacus that can essentially do increment by 1 only? As operand become larger (e.g. $100+100$, $1000*1000$, ...) their abacus becomes more inefficient and mental or writing calculations becomes much efficient. Do they really think that the operations in optical abacus presented here is “wise”, “cool” or “smart?” Otherwise I believe that such an inefficient calculation machine is not appropriate for Nature Communications.

Our Response: We agree with the referee that we do not present a complete all optical computer in our current work. We note though, that the ancillary devices/circuits needed for the realization of an automated system can be envisaged on chip as well, for example by using co-integration with active elements, i.e. through photonic foundry processing. In the revised text we have made clear that our PC-cells can be used as a tool to carry out arithmetic operations, but only the heart of the system is so far implemented on chip and surrounding circuits can be implemented in future work (change to title, introduction, conclusions, see page 4):

“We operate our on-chip photonic circuits at telecom C-band wavelengths and use picosecond optical pulses, that here for simplicity are electronically generated off-chip, to switch the PCM-cells. We note

that in principle the pulse routing and generation could also be integrated on-chip using heterogeneous integration with light sources and through combination with active elements available in photonic foundries [19], [20].”

Moreover, we note that the key idea to perform calculations by analogy to an abacus, as described in the manuscript, does not mean that our system is a simple incremental counter. Indeed, since all place values (ones, tens, hundreds etc. in base-10) are represented by separate, individual PCM-cells, we can manipulate (i.e. calculate with) any power of ten directly. Because we can work directly in base-10, our approach offers a substantial advantage, in terms of computational efficiency, compared to systems operating binary only.

It is true, as pointed out correctly by the reviewer, that we can not move several beads in a single step. However, moving a bead is initiated by a single picosecond pulse and, as explained above, since the crystallization and cooling time of the PCM-cell is the sub-ns range [see also C. Ríos et al., „Integrated all-photonic nonvolatile multi-level memory“, Nature Photonics 9, 725 (2015)], the maximum time needed for moving all (ten) beads of a certain place value would still be less than ten nanoseconds.

Also, as we mention in our response to point #2, since individual PCM-cells represent different powers of ten (1s, 10s, 100s etc), arithmetic calculations are not performed simply by incrementing by ‘1’, but by incrementing in 1s, 10s, 100s, 1000s etc. (so, by way of an example, subtracting 17 from 74 requires only 15 pulses, as explained below).

Additionally we emphasize, that in our PCM-cells the results of a computation are stored in the same PCM unit which carries out the computation. This way no time or energy is lost by moving results to a separate memory as it is necessary in modern computer architectures.

(2) Their PCM cells are not automated for multi-digit calculation and carryover and therefore do not work as computer/processor. They did not report computing at all and so the work/framework should not be compared with computer and/or computer architectures. The current title is invalid. Their framework is not related with non-von Neumann computer architectures.

Our Response: We thank the reviewer for bringing up this point. Therefore we have changed the title and wording to remove claims about optical computing.

(3) The paper claims that PCM cells successfully worked as optical abacus. But I think that the demonstration is at least insufficient. It is obvious that $M*N$ multiplication is emulated by $M*N$ times increment by 1, which should not be an implementation of multiplication for a processing unit. As for subtraction, conversion between decimal number and 9’s complement is needed. How do they

implement the conversion on their PCM cells? It seems that implementation of subtraction (9-7 and 9-9 here) seems difficult for their PCS cells. For computation, how the machine recognizes that which PCM cell records decimal value or its 9's complement. Moreover, even in addition, addend N is replaced by N times addition of 1. After all, what they demonstrated here was just an optical counter as described in Section 5 of Supplementary materials. The fact that $10+10$ ($10*2$) were emulated by $(1+1+1+1+1+1+1+1+1+1+1+1+1+1+1+1+1+1+1+1)$ seems funny. I don't think that this is the demonstration of multistate compute operation. As pointed, such an optical counter is much inefficient than mental calculation. I believe that their current implementation spoils merits of multistate operation and multistate recording. More efficient implementation should be presented as 2-digit addition, subtraction, and multiplication operations can be operated by a few steps.

Our Response: We thank the reviewer for bringing up the point that the description of various 'computing' aspects of our approach were not described clearly enough in the original manuscript. We have adjusted the manuscript accordingly to clarify the benefits of our approach. As already pointed out in our response to (1) above, our abacus is capable of more than just incrementing. For example, calculating $M*N$ is not implemented simply by incrementing by '1' $M*N$ times. Instead, the calculation is performed by adding up N, M times, by making use of more than one digit (i.e. using more than only one PCM-element) – for example, calculating $2*10$ would only take two pulses applied to the PCM-cell representing 10s (not twenty pulses applied to the cell representing 1s).

To show the power of directly operating in base ten and to make clear that our proposed PC-cell is more than a simple counter we moved the original subtraction example from the main text to the supplementary materials. Instead, as suggested by the referee, we added a new figure (Figure 3 in the new manuscript) and associated text in the revised manuscript demonstrating subtraction performed with two PCM-cells, including also implementation of the carryover operation.

We have updated the main text accordingly and included the following passages:

“[...] Figure 3B now shows a subtraction example carried out directly in base-10 and with two digits. Here we use the numbers complements approach which requires addition routines only [20]. We subtract '17' from '74' (which of course should yield the answer '57') by adding '17' to the nine's complement of the minuend '74', which is '25'. In a first step, we set the state of the two PCM-cells corresponding to the minuend. Because we use the nine's complement we switch the first cell (ones) to '5' and the second cell (tens) to '2', so that the phase states of the two elements now hold the value '25'. Now the subtrahend '17' is added, consisting of a single crystallization pulse in the second cell and seven pulses sent to the first cell. After five of these seven pulses, the 'full' crystalline level in the first cell is reached and storage of the carryover is performed. The reset pulse for cell one is now utilized to induce a crystallization step in the second cell (representing the tens), as described above.

After sending all the required pulses, as above, the phase states of the PCM-cells represent the value '42' which is the nine's complement of '57', the correct result of the subtraction."

Regarding the nine's complement, we would like to argue that the PCM-cell itself does not need to know if it performs calculation in 9s complement or not, because it simply executes the corresponding addition task. In a complete all-optical processor the information if a certain value represents the 9s complement could be indicated by an additional PCM-element and would be implemented in the surrounding circuitry.

In addition to the above additions and clarifications in the revised text, we have, in order to avoid any potential confusion in the readers' minds as to the computing architecture content of our work, reformulated the title of our manuscript to "Calculating with light using a chipscale all-optical abacus".

(4) The second part reports addressing and random access to arrayed PCM cells. Although abacus is not a computer, such technology may be needed for multi-digit calculation. However, in Fig. 4, 3*3 PCM array was operated in base-2 (binary) only. Compatibility with the base-10 (decimal) optical abacus in the first part is not demonstrated. In the write (switch) operation (Fig. 3D) it was ambiguous that it was compatible with base-10.

Our Response: The reviewer is correct that multilevel readout in the previous manuscript was not explicitly shown. Therefore we have added a new section in the supplementary information showing that also the readout can be performed with multiple levels. The new section 10, page 10 in the SI with figure S9, is composed as follows:

"10. Multilevel readout in two-pulse mode

Using the waveguide crossing array does not allow for the same simple readout mechanism of the cell states in a transmission measurement which was used for readout of a single cell. Therefore, here a destructive readout scheme is applied that utilizes the change in optical transmission, when a reset pulse is send to the cell. In Fig. S9 five distinct levels were prepared before sending a reset pulse (consisting of two overlapping pulses) to switch the PCM-cell to its initial amorphous phase ('level 0'). By measuring the change in the optical transmission (c1 to c4) all different levels can be distinguished enabling multilevel operation also in a waveguide crossing array.

Since a reset pulse also deletes the information that was stored in the memory element, every readout event can in principle be accompanied by a subsequent write pulse that resets the initial level again. This way the loss of information inherent to destructive readout could be overcome with only a slightly longer readout time composed of two individual pulses."

The author may continue efforts toward acceptance of this work. If they wish to publish the work as optical abacus, addressing the comments (1)-(3) and excluding the second part is my recommendation. Instead of the second part, they should add theoretical discussion about how their optical abacus can be operated much faster than human's mental calculation without the abacus (~millisecond or faster.) Or they may add experimental result of arithmetic operation with total time in millisecond (although multi-step structure may be ambiguous or noisy.) Unlike the optical memory, speed is highly important for a calculator and as a Nature Communications paper such high impact is needed.

Otherwise, if they wish to publish the work as a prototype of all-optical computation, they should withdraw the claim of optical abacus and the topic should be changed to such as all-optical programmable logic device. It may be possible by excluding the not-automated operations (reset/carryover.)

Our Response: We have addressed questions 1-3 as suggested and performed additional experiments, as included in the main text and the revised supplementary materials. We have removed the claim of “computing” and therefore changed the title to include “calculating”, which we believe is more appropriate.

I describe other comments below.

(5) In Figs. 2, and S3, the signal of the PCM-cell 2 (first digit/blue bead) should be plotted as in S5, otherwise the claim of carryover and 2-digit calculation were false and invalid.

Our Response: Figs. 2 and S3 are intended to explain the basic operating concepts and therefore only the first digit was measured to keep the graphs simple and clear. In order to justify the claim of carryover and two-digit calculation, we added a new example (“ $74-17=57$ ”) to the manuscript. This example now shows a real two-digit operation exceeding a simple incremental counter, because this calculation only needs fifteen pulses in contrast to 42 pulses that would have been needed when only incrementing by one. The overall calculation time is less than 15 ns when considering only the switching speed of the cell. This speed is eventually limited by the crystallization time of the material.

(6) Fig. 2B and 2C report very simple 1-digit calculation ($6+6$, $4*3$) which the readers may not feel that such optical abacus is useful. It is also important to demonstrate that the PCM cell was worked not just an optical counter. I recommend the addition and multiplication operations by 2-digits (e.g. $26+35$, $16*5$) and direct arithmetic operation in the first digit (e.g. $2+3$.) If possible, direct change between two multi-level states such as $0 \rightarrow 6 \rightarrow 0$ corresponding to sliding many beads can be used and be good.

Our Response: We agree with the referee and have therefore updated the main text. The new two-digit subtraction example is thus added to the main text (Fig.3). We note that shifting more than one bead at a time is not possible without changing the pulse energies. Yet because shifting a single bead (inducing one crystallization step) takes less than a nanosecond, even shifting ten beads would be faster than ten nanoseconds. To further speed up the calculation, the first addend could always be set using a single pulse per digit, if the pulse energies are adjusted corresponding to the crystallisation level as shown in [C. Ríos et al., „Integrated all-photonic nonvolatile multi-level memory“, Nature Photonics 9, 725 (2015)]. The advantage of only needing two different pulse energies (one for amorphisation and one for a crystallization step) as proposed in the manuscript would then of course be invalid.

(7) According to the text, the addressing of the PCM cells in the waveguide array was achieved by sophisticated techniques combining two-pulse launching, wavelength-division multiplexing and mode-division multiplexing. But detail experimental conditions for Figs. 3 and 4 are not disclosed. More detailed clarifications should be made as someone can reproduce/simulate the experiment.

Our Response: In Figs. 3 and 4 the addressing of the waveguide array was performed off chip. This could in principle be achieved by for example wavelength-division multiplexing. In order to clarify this point, we revised this section in the manuscript. The setup used for the two-pulse switching experiments is shown in Fig. S1, page 3 in the SI.

“Fast on-chip selection of the two waveguides could for example be carried out by wavelength [22] and mode-division [23] multiplexing schemes, e.g. by adjusting the wavelengths and/or modes of the excitation pulses accordingly (note that in this work the selection of the waveguides was realized off-chip and the experimental setup is shown in Fig. S1).”

(8) The random access readout reported in Fig. 4 reset the referred cell to 0. After the random readout, some cells are reset to 0 and other cells hold the data. For instance, 199999999 is changed to 190990999 after the random readout of 2 cells. Do they think such random readout scheme is actually useful?

Our Response: We agree with the referee that a non-destructive readout scheme would be preferable. Because the waveguide crossing architecture used in our experiments does not allow for a simple transmission measurement to detect the cell state as for single PC-cells, we chose destructive readout instead. In order to maintain the cell state, to refresh the old memory state a write pulse after the read-out could be used to set the cell back to its initial state. A readout operation would in this case always consist of a reset and a write pulse and could still be executed in less than a nanosecond. This is now also explicitly stated in the new section of the supplementary information “10. Multilevel readout in

two-pulse mode“ and figure S9. We also point out that some existing memory technologies, such as ferroelectric RAM, also have a destructive readout process, so such a requirement is not per se a barrier to implementation.

As also indicated in the revised manuscript one might in future think of exploiting the post-excitation transient response of PCMs ([27]) to realize a non destructive readout.

(9) The use of following words should be clarified or justified: non-von Neumann computer architectures, scalable, chip-scale, telecom C-band, Turing-complete computers. What they mean random access is ambiguous.

Our Response: We have clarified the nomenclature in the revised manuscript and use the terms as follows:

- By non-von Neumann computing we refer to computing schemes where processing unit and memory storage are not separate entities.
- Random access means, that all cells can be individually addressed, in contrast to memories, where always a bunch of cells must be read, or written.

(10) Fluctuation of the expressions such as abacus/counter, abacus/processor, and abacus/memory makes the paper confusing. They should revise the paper as is more coherent.

Our Response: We have carefully checked the language for consistency and updated the supplementary information and main text accordingly.

We hope to have addressed all concerns of the referee with the additional experimental data, the changes to title, main text and the supplementary materials. We hope that she/he will support publication of our results in Nature Communications.

Reviewer #2 (Remarks to the Author):

I think there is some nice technical work described in this manuscript, in particular the use of PCM cells organized in arrays. So that should be published somewhere.

However, in my opinion the article is somewhat misleading. The title "Computing with light using a chip-scale all-optical abacus", suggests to me that all the arithmetic operations are performed optically. Reading the article and supplementary material though, I get the impression that when a carry-over occurs then a reset pulse must be sent to one digit, and an increment by one pulse sent to the next higher digit. These pulses appear to be generated electronically in the experimental setup.

For the system to be truly all optical there needs to be all optical switching of light pulses by the PCM cells. So that whenever a carry over needs to occur then the optical chip itself can sort out what needs to be done. Since all-optical switching does not appear to occur on the chip presented here I feel that the work does not represent a really significant advance in optical computing that would be needed to warrant publication in Nature Communications.

Our Response: The reviewer is correct in so much as the presented PCM-cell array is not yet an all optical computer, because the picosecond pulses from the femtosecond laser are picked electronically off-chip using an acousto-optical modulator and a pulse generator. Nevertheless, the heart of the calculating system – the waveguide-crossing array that performs the arithmetic tasks is operated all-optically, which we feel is a significant advance by providing optical random access to a chipscale PCM cell array. We note that also the detection of a carryover and the generation of the reset pulse could be done optically, for example by making use of optical bistability in a cavity (for example an add-drop filter configuration). If the input power of the cavity would be small no light would pass to the output port, but if a certain threshold is reached, the light at the output could be used to perform a reset. The threshold in this example corresponds to the transmission level of the crystalline state.

To further address the concerns of the reviewer we revised the manuscript to make clear which parts already work all-optically and which do not. We also changed the title to remove claims about computing, as also suggested by referee #1. In addition, we further point out what the significant advances of our current work are:

- i) The phase-change cell, for the first time, shows convenient stepwise calculations in base ten with an integrated photonic device that can be operated with GHz frequencies.
- ii) The results of such calculations are simultaneously stored by the same integrated photonic cell that carried them out.
- iii) Embedding these cells in the waveguide crossing structure enables a simple selective addressing scheme for individual cells without the need for many laser sources.

The presented structure shows the heart of an optical computer only, but demonstrates that this can work quite well.

- iv) Although the switching circuitry for the incoming pulses and the array is not implemented on chip for experimental reasons (in our setup the switching circuitry is controlled electronically), it could in the future also be performed optically.

In the revised manuscript we included further experimental results, showing subtraction in the nine-complement. The new data also includes the use of the carryover with two separate PCM cells (figure 3 in the main text). We further performed endurance cycling to test the reliability of the switching process, as included in the revised supplementary material (section 11 and figure S10). During more than a million switching cycles we do not observe degradation of the devices, thus illustrating that the operation of our system is robust. Further data is also provided on multi-level readout in two-pulse mode in the SI (section 10 and figure S9).

We hope that with the additional experimental data, the revised title and further changes to the main text and the supplementary materials we have addressed the concerns of the referee. We hope that she/he will support publication of the results in Nature Communications.

Reviewer #3 (Remarks to the Author):

This is an interesting paper, and I would recommend publication in Nature communications after major revision. This paper is built on and an improvement upon the work done previously by the authors. It is effectively a combination of the devices and concepts presented in:

- Integrated all-photonic non-volatile multi-level memory Nat. Photonics 9, 725-732 (2015) and
- Arithmetic and biological inspired computing using phase change materials Adv. Mater. 2011, 23, 3408–3413

Abacus application is weak. I fail to see how this is in anyway an efficient or sensible way of arithmetic computing. In the least the authors should put the abacus in context with alternative technologies and discuss what this adds to the field.

Our Response: We thank the reviewer for her/his support and the helpful suggestions how to improve the manuscript.

We note that the concept of the abacus has to be viewed as a helpful tool for arithmetic calculations that in the current work is used to explain the operation principle of our phase-change cells. Besides the advantages of operating in base ten, the PCM-array (or in analogy the abacus) alone is not an all-optical computing device because it needs an operator who decides which beads to shift. But as already indicated in the answer to reviewer #2, the ancillary circuits needed to perform full all-optical calculations can be envisaged and be built in the future with additional on-chip integration. In order to avoid confusion, however, we have changed the title to reflect that our device is used for calculating and not a full-fledged computer.

The novelty of the manuscript lies in the positive steps made in integrated all-optical computing, the manuscript effectively implements a random access memory integrated photonic cross-bar architecture reminiscent of the recent electronic cross-bar architectures. Operating at telecom c-band wavelengths with ps optical pulses also makes it industrially friendly and comparable to say electronic DRAM speeds. An improved novel two pulse random access scheme is used allowing access to any cell in a phase change array. One major shortfall is that the read operation complementing the two pulse random access scheme is destructive. Effectively making the system a Read-once memory system.

Our Response: We agree with the reviewer that the destructive readout scheme is not ideal. To overcome destructive readout, a two-pulse scheme could be used to directly rewrite the cell after reading. This way the memory system would keep all the information also after the readout. More details regarding the readout process are now given in the supplementary materials of the revised manuscript. We have included the new section 10 and figure S9, where we show multilevel readout in

two-pulse mode. We also point out that some existing memory technologies, such as ferroelectric RAM, also have a destructive readout process, so such a requirement is not per se a barrier to implementation.

In the first half of the manuscript AIST is used as the phase change material. For the random access section the material is changed to GST. Authors should provide the reasoning for this change in the manuscript. How did the pulsing parameters (energy, width) change when using GST? Was AIST not good for random access?

Our Response: The material choices were made for historic reasons related to availability of sputter targets during device fabrication, but do not make a significant difference for the operation of the devices. In our experiments we did not observe any significant differences between GST and AIST when switching the PCM-cells. The reason for using the different materials was only the availability of the respective chips. To show that the arithmetic operations work the same way with GST and AIST, we included new experimental data in the revised main text. The two-digit example added to the manuscript is based on a GST cell (figure 3). Also note that the pulse energies for both experiments are below 20 pJ. We have updated the main text accordingly (page 7, 8):

“(Note that in this second set of experiments we use, for convenience, the phase change material Ge₂Sb₂Te₅, but we do not find significant differences between devices made from AIST or GST and thus are able to maintain the same measurement and fabrication approaches with either material). One pulse is used to provide the row address, while the second pulse yields the column address in our 2D array.”

In addition, we provide a direct comparison between AIST and GST PCM cells in the supplementary materials, new section 3 and figure S2. The two material platforms show very similar behaviour in terms of thermo-optical response, switching energy and switching contrast.

Some points that need to be addressed:

- The authors provide results for random access operation for up to a 3 x 3 array. How does this scale? How does their pulse energies scale with a growing array. A discussion on scaling should be provided.

Our response: Scalability is indeed an important factor, because the device operation is absorption limited. Smaller PCM-elements would allow increasing the array size, but would at the same time reduce the switching contrast. Using four pulses instead of two would also allow to operate larger arrays (eg. to a size of 10x10). Several arrays of these 10x10 matrices can then be joined on chip employing signal amplification between the blocks.

Improvements in device footprint can readily be implemented. The distance between two crossing points in our current layout is about 20 μm but could be reduced to less than 2 μm when utilizing periodic structures and multimode Bloch waves [see Liu et al, Optics letters, “Ultra-low-loss CMOS-compatible waveguide crossing arrays based on multimode Bloch waves and imaginary coupling”, vol 39, no. 2, 2014]. Such an approach would also reduce the insertion loss of the crossings by a factor of ten and may allow larger array sizes.

- Nothing on endurance cycling is provided. How many times do they switch the cells before they fail. Also any difference between single pulse and two pulse scheme in terms of endurance of the cells. As the cells need to be destructively read in the case of the two pulse scheme, this must reduce their endurance significantly?

Our Response: Endurance of PCM devices up to 10^{12} has been shown already, while cycling up to 10^{15} was predicted theoretically. Switching to intermediate states instead of going the whole way from fully amorphous to fully crystalline in a single step should also prevent the PCM elements from failing earlier because lower pulse energies are used.

To show that the two-pulse switching does not affect the cyclability we added a new section about endurance cycling in the supplementary materials (section 11, figure S10). We performed cycling tests with more than 1 million switching cycles without noting degradation of the device. Switching cycles were implemented both with single- and two-pulse switching. We do not find that the cells fail earlier when the two-pulse scheme is applied because the total pulse energies supplied are approximately the same as during single-pulse switching. We have updated the supplementary materials accordingly.

Of course adjustments have most certainly to be made to suppress device failure mechanisms when going to higher cycle numbers, but these would not affect the device operation principle but only the exact structure of the PCM-cell (kind of capping layer, its thickness, exact material composition, etc.).

- I would suggest to move the discussion of “storing the carryover” from supporting info to main text as the splitting of the signal and utilising a high power reset pulses to partially crystallise the second pcm cell was not clear at all in the main text.

- Annotation for figure 1b would be useful to the reader.

Our Response: We thank the reviewer for this suggestion. To point out the computational power of operating in base ten an experimental example using two-digits was added to the manuscript. This now also includes the discussion of the carryover.

Annotations for figure 1B have been added. The main text has been updated with the following sections:

“As already mentioned before the carryover is stored in a second PCM-cell and arithmetic operations with more than a single digit can be performed utilizing the whole computational power of operating directly in base ten. Each cell represents a single place value (ones and tens) corresponding to the different rods of an abacus. Figure 3A shows the experimental pump-probe setup used for two-digit operations including the carryover. To detect the cell-states a continuous wave probe laser is used and the optical transmission through the PCM-cell is monitored with a photodetector. As in the experiments before the pump pulse to switch the cells is picked from a picosecond-laser but the pulse which was formerly used to operate the first phase-change element is now split in two (off-chip, but could in future also be carried out on-chip) and guided to both PCM-cells. If the splitting ratio is adjusted in an appropriate way (using the amplifiers), the part of the energy of the crystallization pulse for the first element which is send to the second cell does not induce a transition. In contrast, if the high energy amorphization pulse is send to the first element, a crystallization step is induced in the second cell, to directly store a carryover.”

We hope to have addressed all concerns of the referee with the additional experimental data, the changes to title, main text and the supplementary materials. We hope that she/he will support publication of our results in Nature Communications.

REVIEWERS' COMMENTS:

Reviewer #1 (Remarks to the Author):

In the revised paper, the authors have address the comments raised by the referees and have made substantial revisions.

Basically, they have made a good job in the revision. Now they withdraw the claim of optical computing/processor in their "abacus" and admit that an abacus needs an operator and some off-chip operations. I think this is a clever choice. In the revision, they added new experimental data. In Fig. 3, simultaneous operation of two digits, carryover and reset made by one pulse, and direct addition of the second (tens) digit ($2+1$) were demonstrated. In supplemental materials, they add detail characteristics of AIST and GST, multi-level readout in two-pulse mode, and endurance test.

Now feasibility of two-digit optical abacus and all-optical multi-level arithmetic operation were unambiguously demonstrated. If multi-sliding of the beads or arithmetic operation at one step ($9+9$ by a few pulses) are realized in future, it will be a candidate of a central element of an optical processor. This work is a cutting-edge usage of phase-changed devices and the technical level is very high.

I think that the paper is now worth consideration for publication in Nature Communications.

I describe additional comments below.

*I think that the mention of (single-taped) Turing machine seems negative considering recent significant advance in computation (such as deep-learning approach and quantum computation.) As for the claim of non-von Neumann arithmetic, this work only demonstrated arithmetic operation in a memory unit and there is no solution for solving the von Neumann bottleneck (do you think that a multi-level flush memory that can perform similar operations electrically is recognized as a non-von Neumann arithmetic?) I think that what is more important in their "chip-scale" optical network is the devices are reconfigurable, namely can be operated as memory, switch, and calculation unit. They may claim that their optical circuit is a first step for optical FPGA.

*The expressions "optical abacus" and "photonic abacus" may be unified.

*The technical term "telecom C-band" should be clarified.

Reviewer #3 (Remarks to the Author):

The authors have addressed all the points raised in a satisfactory manner. I would recommend publication in Nature Communications.

Calculating with light using a chip-scale all-optical abacus

J. Feldmann, M. Stegmaier, N. Gruhler, C. Ríos, H. Bhaskaran, C.D. Wright, W.H.P. Pernice

Response to reviewers

We thank all the referees for their insightful reviews of our manuscript and their very helpful recommendations on how to improve it. In the following we describe the changes made in order to address the reviewers comments.

Response to reviewer 1:

In the revised paper, the authors have address the comments raised by the referees and have made substantial revisions.

Basically, they have made a good job in the revision. Now they withdraw the claim of optical computing/processor in their “abacus” and admit that an abacus needs an operator and some off-chip operations. I think this is a clever choice. In the revision, they added new experimental data. In Fig. 3, simultaneous operation of two digits, carryover and reset made by one pulse, and direct addition of the second (tens) digit ($2+1$) were demonstrated. In supplemental materials, they add detail characteristics of AIST and GST, multi-level readout in two-pulse mode, and endurance test. Now feasibility of two-digit optical abacus and all-optical multi-level arithmetic operation were unambiguously demonstrated. If multi-sliding of the beads or arithmetic operation at one step ($9+9$ by a few pulses) are realized in future, it will be a candidate of a central element of an optical processor. This work is a cutting-edge usage of phase-changed devices and the technical level is very high. I think that the paper is now worth consideration for publication in Nature Communications.

I describe additional comments below.

*I think that the mention of (single-taped) Turing machine seems negative considering recent significant advance in computation (such as deep-learning approach and quantum computation.) As for the claim of non-von Neumann arithmetic, this work only demonstrated arithmetic operation in a memory unit and there is no solution for solving the von Neumann bottleneck (do you think that a multi-level flash memory that can perform similar operations electrically is recognized as a non-von Neumann arithmetic?) I think that what is more important in their “chip-scale” optical network is the devices are reconfigurable, namely can be operated as memory, switch, and calculation unit. They may claim that their optical circuit is a first step for optical FPGA.

Our Response: We again thank the reviewer for his helpful comments on how to further improve the manuscript. Following his suggestions, we rewrote the introduction and removed the reference to the “single-taped Turing machine”. Also, the advice to compare our devices to electrical FPGAs is now taken in to account and added at the end of the introduction.

“Introduction

Virtually everyone is familiar with the abacus, probably invented by the Sumerians between 2700 and 2300 BC and one of the earliest mathematical tools¹. In its most widely known form, the abacus consists of a frame, rods (or wires), and beads to implement a mechanical multistate machine. Each rod represents a different place value (ones, tens, hundreds, etc.), while each bead represents a single digit. By sliding the beads along the rods in appropriate ways, all the basic arithmetic functions of addition, subtraction, multiplication, and division can be carried out, along with even more complex operations. At the same time, the abacus stores the result of such calculations in the (final) position of its beads. In essence, the abacus provides two of the most basic functions of a computer, namely processing (calculation) and memory (storage), and it does this simultaneously and in a single device (or, as an alternative description, at one and the same location). Modern computer systems however, based as they are on the so-called von Neumann architecture, separate, in time and space, the operations of processing and memory. Processing is carried out in the central processing unit (CPU), while separate memory devices store the results of any calculations carried out by the CPU. The constant transfer of data between CPU and memory leads to a ‘bottleneck’ in terms of the overall speed of operation (the well-known von Neumann bottleneck) and wastes very significant amounts of energy. Computer architectures that can somehow fuse together the two basic tasks of processing and memory (i.e. non-von Neumann architectures) therefore offer tantalizing potential improvements in terms of speed and power consumption. The search for such new computing approaches has been boosted by the advent of so-called memristive devices, i.e. devices that can be excited into multiple (non-volatile) states and whose current state depends on their past history²⁻⁴. Indeed, such memristive devices can both process and store data simultaneously, and have led to the new concept of multistate *memprocessor* or *memcomputer* machines that compute with and in memory^{5,6}. These new approaches to computation provide not only the same computational power as a universal Turing machine (describing all conventional digital computers), but also a range of additional and attractive properties including intrinsic parallelism, learning and adaptive capabilities and, of course, the simultaneous execution of processing and storage that removes the need for continual transfers of data between a CPU and external memory⁵. Computer architectures based on the multistate compute-and-store

operation of a simple abacus can also provide us with such a radically new approach to computing, and one that can work directly in high-order bases rather than just binary. Carrying out such a radical approach entirely in the optical domain using integrated chip-scale photonics would allow for exploiting the ultra-fast signaling and ultra-high bandwidth capabilities intrinsic to light⁷.

In this work we demonstrate a key component in this quest, namely an integrated all-optical abacus-like arithmetic calculating unit. Our approach is based on the progressive crystallization of nanoscale phase-change materials (PCMs) embedded with nanophotonic waveguides. PCMs have been the subject of intense research and development in recent decades, mainly in the context of rewritable optical disks and non-volatile electronic memories⁸⁻¹⁰. A key feature for such applications is the high contrast in both the electrical (resistivity) and optical (refractive index) properties of PCMs between their amorphous and crystalline states⁸⁻¹⁰. The high refractive index contrast means that if we embed PCMs into nanophotonic waveguides we can switch them between states using optical pulses sent down the waveguide, and readout the resulting state optically too. Previous work has used such an approach to demonstrate integrated all-optical memories^{11,12}. Here we show that *processing* and storage is in fact possible, demonstrating an on-chip abacus-like photonic device that simultaneously combines calculation and memory. We carry out base-10 additions and subtractions (including carry-over) in a single PCM cell using picosecond optical pulses and energies in the nanojoule range. We also demonstrate successful random user-selective access to each PCM-cell in a two-dimensional array. Moreover, our approach is scalable and could provide photonic integrated circuits with devices that are reconfigurable, namely can be operated as memory, switches, and calculation units, perhaps providing the first steps towards the optical equivalent of electronic field-programmable gate arrays (FPGAs).”

*The expressions “optical abacus” and “photonic abacus” may be unified.

*The technical term “telecom C-band” should be clarified.

Our Response: To unify the expressions, the term “photonic abacus” is now used throughout the whole text. The wavelength range corresponding to the “telecom C-band” was added for clarity.

Reviewer #3 (Remarks to the Author):

The authors have addressed all the points raised in a satisfactory manner. I would recommend publication in Nature Communications.

Our Response: We thank the reviewer for her/his support of our manuscript and we are delighted that she/he shares his/her excitement about our work.